# Physiological Characteristics and Transcriptomic Responses of *Pinus yunnanensis* Lateral Branching to Different Shading Environments

**DOI:** 10.3390/plants13121588

**Published:** 2024-06-07

**Authors:** Chiyu Zhou, Xuesha Gu, Jiangfei Li, Xin Su, Shi Chen, Junrong Tang, Lin Chen, Nianhui Cai, Yulan Xu

**Affiliations:** 1Key Laboratory of Forest Resources Conservation and Utilization in the Southwest Mountains of China, Ministry of Education, Southwest Forestry University, Kunming 650224, China; zhouchiyu@swfu.edu.cn (C.Z.); 18283457310@163.com (X.G.); ljfei9661@126.com (J.L.); suxin@swfu.edu.cn (X.S.); chenshi@swfu.edu.cn (S.C.); tjrzy2016@swfu.edu.cn (J.T.); linchen@swfu.edu.cn (L.C.); cainianhui@swfu.edu.cn (N.C.); 2Key Laboratory of National Forestry and Grassland Administration on Biodiversity Conservation in Southwest China, Southwest Forestry University, Kunming 650224, China

**Keywords:** Yunnan pine (*Pinus yunnanensis*), the branching ability, shading stress, RNA-seq, physiological characteristics

## Abstract

*Pinus yunnanensis* is an important component of China’s economic development and forest ecosystems. The growth of *P. yunnanensis* seedlings experienced a slow growth phase, which led to a long seedling cultivation period. However, asexual reproduction can ensure the stable inheritance of the superior traits of the mother tree and also shorten the breeding cycle. The quantity and quality of branching significantly impact the cutting reproduction of *P. yunnanensis*, and a shaded environment affects lateral branching growth, development, and photosynthesis. Nonetheless, the physiological characteristics and the level of the transcriptome that underlie the growth of lateral branches of *P. yunnanensis* under shade conditions are still unclear. In our experiment, we subjected annual *P. yunnanensis* seedlings to varying shade intensities (0%, 25%, 50%, 75%) and studied the effects of shading on growth, physiological and biochemical changes, and gene expression in branching. Results from this study show that shading reduces biomass production by inhibiting the branching ability of *P. yunnanensis* seedlings. Due to the regulatory and protective roles of osmotically active substances against environmental stress, the contents of soluble sugars, soluble proteins, photosynthetic pigments, and enzyme activities exhibit varying responses to different shading treatments. Under shading treatment, the contents of phytohormones were altered. Additionally, genes associated with phytohormone signaling and photosynthetic pathways exhibited differential expression. This study established a theoretical foundation for shading regulation of *P. yunnanensis* lateral branch growth and provides scientific evidence for the management of cutting orchards.

## 1. Introduction

Lateral branching is an important component of plant morphological architecture, and it impacts plant light, water, nutrients, and reproductive growth [1,2]. Lateral branching, which reflects adaptive strategies for survival or growth in response to environmental changes, is critical for plant biomass, resource acquisition, and efficient utilization of light [3]. In addition, research has found that the growth of lateral branching is influenced by internal, developmental, and environmental factors [4,5], thereby ensuring branches’ survival under stress conditions. The formation of shoot branching involves a two-stage developmental process [6]. Lateral branch formation occurs in two stages: initially, the axillary meristem within the leaf axil develops into an axillary bud; subsequently, this axillary bud elongates and forms a lateral branch [7]. Classic experiments have demonstrated that the shoot apex suppresses lateral branch growth, a process known as apical dominance. Thus, removing apical dominance benefits branch growth and development. If there is an abundance of nutrients and the environmental conditions are favorable, the axillary bud development will lead to lateral branching [8].

Light intensity provides the essential energy for photosynthesis, influences a range of physiological processes, and is a critical environmental factor for plant growth [9]. In the natural environment, changes in light intensity, duration, and spectral composition alter the light environment and affect plant growth [10]. Under shaded conditions, plants detect changes in the light environment and rapidly trigger a series of morphological adaptations, which is known as the shade avoidance syndrome (SAS) [11]. This response is characterized by hypocotyl and petiole elongation, reduced branching and leaf expansion, and earlier flowering in plants [12,13]. Furthermore, shade stress impacts plant growth, resulting in morphological changes, diminished photosynthetic capacity, and alterations in enzyme activity [14,15,16]. In addition, light has a major influence on morphological and physiological changes in plants [17,18]. In both herbaceous and woody plants, changes in light intensity often lead to alterations in CO_2_ assimilation rates, nonstructural carbohydrates, and phytohormone levels [19]. Under shaded conditions, plants strive to maximize light capture to boost photosynthetic yield, which enhances the rapid fixation of carbon dioxide and the accumulation of carbohydrates. This process helps ensure stable growth conditions for the plant [20]. Important factors affecting plant biomass and metabolism are light intensity and quality.

Light intensity influences plant physiological and morphology characteristics in several ways, such as altering the content of photosynthetic pigments, levels of osmolytes, and concentrations of reactive oxygen species [21,22]. Antioxidant metabolism protects plants from abiotic stress, such as drought, temperature, and shading [23]. Thus, boosting plant antioxidant defenses has the potential to bolster plant resilience against a range of stressors. These defenses comprise key enzymes like peroxidase (POD), catalase (CAT), and superoxide dismutase (SOD) [24]. With increasing shade, photosynthetic pigments and enzymatic activity content respond to varying degrees [25]. Previous studies have reported that shading has been observed to elevate enzyme activity in shade-tolerant tree species, consequently amplifying the effectiveness of lignin synthesis [26,27]. In addition, the level of malondialdehyde (MDA) serves as a crucial marker for lipid peroxidation reactions within cell membranes, allowing for the assessment of oxidative levels to some extent and the interpretation of antioxidant enzyme results [28,29]. Thus, unveiling the response patterns of these enzyme activities to shading will significantly enhance our comprehension of branch shade tolerance.

Phytohormones were crucial components in plant development, evolution, environmental adaptation, and abiotic stress resistance [30]. Plants intricately manage their metabolism by orchestrating a network of phytohormones, enabling them to adjust to variations in both the quantity and quality of light [31]. Shading prompts a complex adjustment in various hormones within plants, including auxins (IAA), cytokinins (CTK), abscisic acid (ABA), and gibberellins (GA). Moreover, research indicates that the biosynthesis of plant IAA and phytochromes is induced under shading conditions [32,33]. Shading in sunflowers alters the endogenous level of IAA, leading to the accumulation of IAA in the internodes and consequently promoting plant elongation [34]. However, shading influences the content of GA, which, in turn, impacts the stability of the PIF/DELLAs complex, inhibiting their interaction [35]. In shaded environments, different concentrations of ABA have varying effects on plant growth, indicating that ABA is involved in regulating plant responses to shading [36,37]. Different lighting conditions variably affect the biosynthesis and transport of hormones within plants. Light exposure can regulate the content of JA in *Arabidopsis*, thereby altering its morphological characteristics [38]. Thus, there is some correlation between hormonal changes in the plant and growth rate, biomass, and stress resistance.

*Pinus yunnanensis* is an important needle-leaved tree species in China, renowned for its strong adaptability and drought tolerance characteristics [39]. However, the growth of *P. yunnanensis* after afforestation is extremely slow, with seedlings primarily in a stunted growth phase in practical forestry production. The prolonged seedling period significantly reduced afforestation efficiency, making it difficult to fully realize the ecological value of *P. yunnanensis*. With the continuous advancement of theories and techniques in selecting and breeding superior *P. yunnanensis* varieties, high-quality *P. yunnanensis* families and clones have become necessary germplasm resources for seedling cultivation. Asexual propagation not only ensures the stable inheritance of superior traits from the mother tree but also shortens the breeding cycle. In actual production, the mother trees providing cuttings for cutting orchards are mostly younger. During their growth and development, these mother trees are affected by shading. Therefore, it is very important to investigate the effects of shading on the growth and development of lateral branches in *P. yunnanensis*. However, current research in *P. yunnanensis* mainly focuses on nutrient content [40], transcriptome sequencing [41], community structure characteristics [42], and biomass [43]. There was little information available on the effect of different light conditions on the growth of the branches of coniferous trees. Therefore, the regulation of shading on the molecular mechanisms of branch growth in *P. yunnanensis* should be further investigated. In this research, we examined the effects of different shading treatments on the branch number, branch length, biomass, photosynthetic pigment content, enzyme activity, endogenous hormone levels, and transcriptomic profiles in the lateral branch. This comprehensive analysis aimed to elucidate the growth, photosynthetic characteristics, physiological traits, and molecular responses of the branches to shading. The research results provided insights into the molecular responses of *P. yunnanensis* branching to shading, laying a foundation for the scientific management of cutting orchards.

## 2. Results

### 2.1. Effect of Shading Treatment on Lateral Branching Ability of P. yunnanensis Seedlings

The variations in lateral branching numbers under different shading treatments with stress time are shown in Figure 1. The shoot branching number of all the treatments increased firstly, then decreased, and increased again with the shading time, on the whole. From April to December, The CK treatment resulted in significantly more lateral branching than the L1, L2, and L3 shading treatments. The branch length in different treatments exhibited a growth trend corresponding to the duration of shading (Figure 1). The findings indicated that the number of branches under different shading treatments varied over time and that shading treatments reduced the branching ability of *P. yunnanensis* seedlings.

### 2.2. Response of Biomass Accumulation in Various Organs of P. yunnanensis to Different Shading Intensities

Main and lateral root biomass in CK treatment was significantly higher than in L2 and L3 treatments (*p* < 0.05). The root biomass decreased with the degree of shading after decapitation (Figure 2a,b). Similarly, stem biomass showed a downward degree of shading, with no difference among treatments of CK and L1 (Figure 2c). In contrast, a significant difference was observed when compared with L2 and L3 treatment (*p* < 0.05). The biomass of needles declined with the degree of shading, and CK treatment was significantly higher than L2 and L3 and had no significance with L1 (Figure 2d). Under the CK treatment, the biomass of the lateral branches was higher than that of the other three treatments, and there was a significant difference compared to the heavy shading (L3) treatment (*p* < 0.05). The biomass of the lateral branch initially decreased, subsequently incrementally, and then reduced again with increasing degrees of shading (Figure 2e). The branch biomass under the L1 treatment was higher than that under the CK, L2, and L3 treatments (Figure 2f).

### 2.3. Effect of Shading Treatments on Photosynthesis of P. yunnanensis Seedlings

Different degrees of shading have different effects on chlorophyll content (Figure 3). From June to July, the chlorophyll content in the L1 treatment was the highest among the four treatment groups (Figure 3a). From mid-July to November, chlorophyll content in the shade treatment group was lower than in the CK treatment group. Between June and October, the chlorophyll b content of CK treatment was higher than that of L1, L2, and L3 treatment (Figure 3b), while that of L1 was higher than CK, L2, and L3 between November and December. The trend of chlorophyll b content in different treatments mirrored that of chlorophyll a. The L1 treatment showed a higher total chlorophyll content than the CK treatment in the early and later periods of shading (Figure 3c). The total chlorophyll and chlorophyll a showed higher content under CK and L1 treatment and lower under L2 and L3 treatments. By the large, in different months, the total chlorophyll content first decreased and then increased, while that of L2 declined between November and December. As Figure 3d presented, the carotenoid of L1 treatment was higher than CK’s in the early and later periods of shading. The monthly variation in carotenoids under four treatments decreased during June and July and increased from July to September, except for the L1 treatment. However, all decreased in December.

### 2.4. Effect of Shading Treatments on the Physiological Characteristics of P. yunnanensis Seedlings

Under different shading treatments, the content of SOD, POD, CAT, and MDA in lateral branching of *P. yunnanensis* seedlings first increased and then reduced with shading time on the whole (Figure 4a–d). In our study, the soluble sugar content under different treatments increased with the shading time (Figure 4e). In contrast, the soluble protein content of all four treatment groups showed a trend of decreasing and then increasing (Figure 4f).

According to the correlation analysis of branch numbers, branch length, the biomass of branch, chlorophyll content, carotenoid, and other six physiology traits of *P. yunnanensis* seedlings, the lateral branching numbers were positively correlated with chlorophyll, carotenoid, soluble sugar, SOD and CAT, negatively correlated with soluble protein, MDA and POD, suggesting that the increase in soluble protein, MDA, and POD activity was not conducive to the lateral branching ability of *P. yunnanensis* (Appendix A). On the contrary, chlorophyll a, chlorophyll b, and carotenoids benefitted the number of branches. The branch length was positively correlated with chlorophyll, carotenoid, soluble sugar, SOD, and CAT, and negatively correlated with soluble protein, MDA, and POD, indicating that soluble protein, MDA, and POD inhibited the growth of lateral branching. The research findings on *P. yunnanensis* clearly demonstrate that the biomass of the branch is positively correlated with chlorophyll, carotenoid, MDA, SOD, and CAT. Conversely, it is negatively correlated with soluble sugar, soluble protein, and POD activity. These results strongly suggest that an increase in soluble sugar, protein, and POD activity can have a detrimental effect on the accumulation of branch biomass. The allocation of biomass in the lateral branching of *P. yunnanensis* showed a negative correlation with chlorophyll, carotenoid, and soluble sugar content, while CAT exhibited a positive correlation with soluble protein, MDA, POD, and SOD. These relationships suggest that chlorophyll content, soluble sugar, and CAT significantly influence the allocation patterns of branching biomass in *P. yunnanensis*. In contrast, soluble protein, MDA, POD, and SOD contributed to it. These results suggest that increasing chlorophyll content, SOD, and CAT activities can boost photosynthetic capacity and safeguard plant cells from oxidative damage, thus promoting the growth of *P. yunnanensis* branches.

### 2.5. Response of Endogenous Hormone to Shading Treatments

The endogenous hormones varied in different trends and degrees of shading (Figure 5). The content of salicylic acid (SA) kept increasing under shading, and that of L3 was higher than CK, L1, and L2; the minimum value was observed in CK treatment. Contrarily, the content of Gibberellins (GA3 and GA4) decreased with the shading degree. Suggesting shading treatments suppressed the accumulation of GA3 and GA4. The levels of abscisic acid (ABA), auxin (IAA), jasmonic acid-isoleucine (JA-Ile), isopentenyl adenine (2iP), and methyl jasmonate (MeJA) displayed a similar trend, initially declining and then increasing. However, the contents of jasmonic acid (JA) and Gibberellins (GA7) in the lateral branching of *P. yunnanensis* first increased, then decreased, and subsequently increased again as the severity of shading intensified. This indicates that varying degrees of shading can enhance the levels of JA and GA7 in the lateral branching of *P. yunnanensis*. The variation in trans-zeatin-riboside (ZR) was similar to that of JA, while that of isopentenyl adenine nucleoside (IPA) tended to increase and decrease. The correlation analysis between the lateral branching biomass, lateral branching biomass allocation, lateral branching number, lateral branching length, and endogenous hormone showed that lateral branching biomass negatively correlated with IAA, ABA, GA7, JA-Ile, 2IP and positively correlated with SA, GA3, JA, GA4, ZR, IPA, MeJA (Appendix A). Branch biomass allocation was negatively correlated with GA3, GA7, GA4, ZR, IPA, and MeJA and positively correlated with SA, IAA, ABA, JA, and JA-Ile. Lateral branching number negatively correlated with SA, ABA, JA, JA-Ile and positively correlated with GA3, IAA, GA7, GA4, ZR, 2IP, IPA, and MeJA. In terms of lateral branching length, it negatively correlated with IAA, ABA, JA, JA-Ile, and significantly with the SA; meanwhile, lateral branching length positively correlated with GA3, GA7, ZR, 2IP, IPA, and MeJA, especially the content of GA4. The increase in SA, ABA, JA, and JA-Ile limited the lateral branching number and lateral branching length but promoted the biomass allocation patterns of lateral branching. The increase in GA3, IAA, GA7, GA4, ZR, 2IP, IPA, and MeJA levels facilitated the growth of lateral branching and biomass accumulation, while the concentration of GA4 directly affected the lateral branching length.

### 2.6. Transcriptomic Sequencing and De Novo Assembly and Annotation

Overall, we obtained 68.29 Gb of clean data from the transcriptome sequencing of 12 samples. We conducted assembly and statistical analysis of the raw data of each sample (Appendix A). After quality control, the Q20 for all 12 samples was greater than 97.54%, and the Q30 was above 93.59%. The 12 sequencing samples yielded a total of 511,861,556 raw reads. Having filtered and checked the raw data, a total of 455,291,990 clean reads were obtained. The final transcriptome data obtained are reliable and suitable for subsequent analysis of DEGs.

Then, the uni-genes function annotation was performed. A total of 5708 (10.09%) uni-genes were matched to a sequence in the specified databases. According to an analysis using the GO (Gene Ontology) database, 55 terms were identified as being significantly represented: the category ‘biological process’ contained the highest number of terms, with 28, followed by ‘molecular function’ with 24 terms, and ‘cellular component’ with 3 terms (Figure 6). By aligning the gene sequences with the Nr database, it was found that the sequences showed the most significant matches with gene sequences from *Picea sitchensis* (40.23%), followed by *Amborella trichopoda* (5.22%), *Quercus suber* (3.04%), *Pinus taeda* (2.53%), *Nelumbo nucifera* (2.29%), *Pinus tabuliformis* (2.03%), and *Cinnamomum micranthum f. kanehirae* (1.9%) (Appendix A).

### 2.7. Analysis of Differentially Expressed Genes in P. yunnanensis under Different Shadings

Firstly, The Venn diagram showed that there were no common DEGs among the four treatment groups (Figure 7A). As seen in Figure 7B, there were more DEGs in the CK vs. L3 comparison group, which included 1838 DEGs with 1014 upregulated and 824 downregulated. In the CK vs. L2 comparison group, there were 1238 DEGs, comprising 726 upregulated and 512 downregulated. The CK vs. L1 comparison group featured 801 DEGs, with 595 upregulated and 206 downregulated. In the L1 vs. L3 comparison group, there were 362 DEGs, with 226 upregulated and 136 downregulated. Finally, in the L1 vs. L2 comparison group, there were 266 DEGs, with 99 upregulated and 167 downregulated. In addition, hierarchical clustering was performed (Figure 7C). The genes in the branches of *P. yunnanensis* may be regulated by shading.

The role of genes in the growth of *P. yunnanensis* branches can be studied through KEGG pathway enrichment analysis of DEGs. The top 20 KEGG pathways were significantly enriched in differentially expressed genes under various shading conditions in *P. yunnanensis* (Figure 8A–F). Metabolic pathways significantly enriched in differentially expressed genes across four comparison groups include “Plant hormone signal transduction”, “Phenylpropanoid biosynthesis”, “Starch and sucrose metabolism”, “Photosynthesis”, “Fructose and mannose metabolism”, and “MAPK signaling pathway-plant”. The results unequivocally demonstrate that the pathways mentioned above are crucial in the growth differentiation of *P. yunnanensis* branches in various shade treatments.

### 2.8. Response of Related Genes in the Photosynthesis Pathway to Different Shade Treatments

In the KEGG pathway analysis, the photosynthesis pathway is comprised of four distinct parts: Photosystem I; Photosystem II; cytochrome complexes; and ATP synthase (Figure 9). In the current investigation, we assessed the transcript abundance of four gene categories (Figure 9): photosystem I reaction center subunit (*Psa*); photosystem II reaction center subunit (*Psb*); Cytochrome b6-f complex (*Pet*); and ATP synthase gamma (*F-type ATPase*). Expression levels of genes in the photosynthesis pathway showed a downregulation with deeper shade intensity. In total, we identified 6 *Psa* genes (*PsaD*, *PsaG*, *PsaH*, *PsaL*, *PsaE*, *PsaF*), all of which showed upregulation in the CK condition compared to L1, L2, and L3. Furthermore, the following twelve common DEGs associated with the photosynthesis pathway were identified: *PsbP*; *PsbR*; *PsbS*; *Psb2*7; *PsbO*; *ATPase-gamma*; *ATPase-a*; *PetE*; *PetF2*; *PetF*; *PetH*; *PetC* in different comparison groups.

### 2.9. Response of Related Genes on the Hormone Signal Transduction Pathway to Different Shade Treatments

Phytohormones are important in the growth of *P. yunnanensis* branches, including the regulation of hormone signal transduction and biosynthesis. In the phytohormone signal transduction pathway, a large number of differential genes are enriched in five different hormone pathways (Figure 10). Only one deg was enriched in the growth hormone hormone pathway in four comparisons of different shade levels. The results indicate that the expression of small auxin-up RNA genes (*SAUR*) was upregulated in L3, and the *SAUR* gene exhibited a gradual upregulation with increasing shading intensity. Furthermore, the involvement of histidine-containing phosphotransferase protein (*AHP*) in the cytokinin signal transduction pathway was observed, with downregulation correlating with increasing shading levels. The ABA receptor PYL family (*PYR/PYL*) was upregulated in the ABA biosynthesis pathway and showed high expression in the L1 group. However, the negative regulatory factor, protein phosphatase 2C (*PP2C*), had low expression in the heavy shade (L3). It can be seen that shading affected the expression of *PP2C* in *P. yunnanensis*, and the *PP2C* gene may play a key role in shade inhibition of branch growth. In the JA signaling pathways, all *JAZ* genes were downregulated in CK, so shading might promote high expression of the *JAZ* gene. The SA signal transduction pathway has been found to have three genes encoding pathogenesis-related protein 1 (PR-1) that were identified as DEGs. Notably, all DEGs were discovered to be upregulated in the CK treatment. One deg (*TGA*) was upregulated in the four shading treatments. It is possible that changes in the expression levels of genes on the SA signal transduction pathway are related to the growth and development of *P. yunnanensis* branches. In other words, under different shade conditions, the gene expression of the *P. yunnanensis* hormone signal transduction pathway was changed, which may indirectly affect the phytohormone.

### 2.10. RT-qPCR Validation of Differential Genes in P. yunnanensis under Different Shading Levels

We selected 6 DEGs in photosynthesis and phytohormone signaling for RT-qPCR to verify the quality of the transcriptome. The results of this study found that the trend of gene expression in RT-qPCR was basically almost the same as that of the transcriptome (Figure 11). The transcriptome data from the above results were found to be relatively reliable.

## 3. Discussion

### 3.1. Lateral Branching in Response to Different Shading Treatments

Plants are regulated by light to grow and develop, and they can respond to changing light conditions by adjusting their various physiological responses. Decapitation can promote the occurrence of lateral branching of seedlings [44], and light is important in the growth and development of lateral branching [45]. Moreover, plants have demonstrated phenotypic plasticity, allowing them to adapt their growing patterns and structural architecture to light conditions [46]. Consequently, light exposure constitutes a critical environmental factor in the lateral branching growth processes of *P. yunnanensis*. In the present study, the branching of annual Pinus yunnanensis under different shade conditions was studied. The results found that the number of lateral branching of *P. yunnanensis* decreases gradually with the intensification of shading under different shading treatments, the number of lateral branching in the control group (CK) was significantly greater than that in the mild shading (L1), moderate shading (L2), and heavy shading (L3) respectively. This suggests that there is a higher priority given to existing lateral branching growth, which could limit the development of new lateral branching when the plant is subject to light restriction [47]. This study also explains why the length of the branch increases with increasing shade. Proper light intensity constitutes a fundamental precondition for a plant’s regular growth and maturation, and excessive shading adversely impacts plant development, yield, and productivity [48].

Biomass is one of plant’s essential biological and functional characteristics, reflecting the accumulation of plant materials and the ability to use environmental resources [49]. In addition, most plants adapt to the shaded condition by changing the biomass pattern to capture more light energy and improve their survival competitiveness [50,51]. Shading increases biomass allocation to leaves and stems at the expense of roots to maximize leaf area and reduce growth [52]. In this study, the biomass of the main root, lateral root, main stem, and needle showed a decreasing trend with increasing shading. However, the variation tendency of branching biomass was opposite to other plant organs, and the shoot branching biomass was the highest under mild shade treatment. This indicated that moderate shade can promote accumulation. The shade-avoidance response (SAR) is characterized by rapid stem and petiole elongation, needle growth, apical dominance, and reduced root growth [53]. Here, the root biomass of *P. yunnanensis* showed a decreasing trend with the increasing shading levels. Previous studies have shown that soil moisture, shade, and planting methods can stimulate root growth and enhance a species’ invasive ability [54]. Based on these results, we found that the biomass of *P. yunnanensis* may be reduced in the underground part and transferred to the aboveground part with different levels of shading. The results of our research are consistent with those of Chory [55] and Kohnen et al. [36], who reported that plants had remarkable plasticity in their ability to resist these unfavorable conditions and to react with local changes in growth, metabolism, and reproduction in order to adapt optimally to their environment.

### 3.2. Comparison of Photosynthetic Pigment and Physiological Characteristics under Shading

Photosynthesis represents a critical process underpinning plant growth. However, a plant’s ability to achieve high levels of photosynthesis is dependent on its environmental conditions [56]. Chlorophyll is a vital component of plant systems, as it absorbs, transports, and converts light energy [57]. Chlorophyll can influence photosynthetic activity, which increases chlorophyll content and net photosynthetic rate (Pn) [58]. Chlorophyll b stabilizes chlorophyll-binding proteins and helps adapt to different light environments [59]. In addition, plants produce bioactive carotenoids in response to stress [60]. When plants are exposed to the stress of shade, the levels of chlorophyll-binding proteins in the photosynthetic system are increased. This results in the binding of more chlorophyll, which can prevent the degradation of chlorophyll and photooxidation and ultimately improve the chlorophyll content in plants. The increased chlorophyll content allows plants to absorb more light from shaded environments [16]. In this study, the lateral branches of *P. yunnanensis* showed the highest chlorophyll concentration in L1, while the chlorophyll content in L2 and L3 was lower than that of the control and L1. The studies above indicated that mild shade could promote chlorophyll synthesis. The following statement indicates agreement with the research findings of Na et al. [61] for the photosynthetic characteristics of *Pinellia ternate* under different shade conditions.

When plants are affected by abiotic stress, the contents of osmotic substances are usually increased or degraded to adjust [62]. Plant photosynthesis and soluble sugar synthesis, transport, and accumulation are affected to some extent and indirectly affect plant photosynthesis and cell osmotic pressure in shaded conditions [63,64]. In this research, the soluble sugar under each treatment showed a different rising trend. However, in comparison with the control group (CK), the content of soluble sugars in the other treatment groups decreased as the shade increased. This study demonstrates that the growth of *P. yunnanensis* lateral branching is subjected to a certain degree of low-light stress, which affects photosynthesis and inhibits the synthesis of soluble sugars. However, the phenomenon of soluble protein levels increasing with the intensification of shading was observed; this may reflect a compensatory growth response of *P. yunnanensis* lateral branching to low-light conditions. The antioxidant enzyme mechanism is crucial in plants’ defense against various abiotic stresses, including high temperatures, intense light exposure, and drought [65]. Shade stress induces the overproduction of plant reactive oxygen species, resulting in oxidative damage and metabolic disorders that inhibit plant growth [66]. Research indicates that the activity of antioxidant enzymes undergoes varied changes influenced by factors such as cultivar, duration of shading treatment, growth phase, and intensity of shading [67,68]. In this experiment, the levels of SOD, POD, CAT, and MDA in *P. yunnanensis* varied in response to different shading treatments. This may have occurred because plants were unable to resist stress and were shaded beyond their tolerance, which reduced SOD activity [69]. Therefore, the increase in enzyme activity may be due to plants adapting to shading by regulating their enzyme activity to reduce the damage caused by shade stress. This result aligns with that of El-Beltagi et al. [70]. In the shaded environment, the metabolic capacity of *P. yunnanensis* may be compromised, resulting in the continuous generation of reactive oxygen species that exceed the plant’s tolerance range for growth regulation. This scenario results in a reduction in the activities of SOD, POD, and CAT, ultimately leading to the suppression of growth and development in *P. yunnanensis*.

### 3.3. Role of Phytohormones in the Growth and Development of Lateral Branching

Phytohormones have a significant impact on controlling the quantity of lateral branching [71]. Phytohormones are essential in controlling plants’ growth, development, and nutrient distribution, significantly influencing their capacity to adapt to diverse environmental conditions [72]. In response to adversity or stress, plants exhibit changes in the content of various hormones, which regulate the physiological activities of the plant [73]. In the experiment, the contents of GAs (GA_3_, GA_4_, and GA_7_) exhibit a declining trend with the increasing severity of shading. Gibberellin has been shown to act as a positive regulator in the control of lateral branching in a previous report on the woody plant *Jatropha curcas* [74]. The content of JA in each shading treatment group increased to different degrees compared with CK under various shading conditions. The results of this study agree with those previously reported [75]. Cytokinin (CTK) can enhance cell division and break the dormancy of buds, promoting axillary bud growth and development to lateral branching [76,77]. The phenomenon of shading has been observed to exert an influence on CTK levels, resulting in a suppression of lateral branching growth and development [78]. This indicates that, in specific circumstances, elevated levels of CTK may be more advantageous in regulating the expansion of lateral branching growth. In this study, under the shading conditions, A reduction in the concentrations of ZR, iPA, and 2IP was identified as a factor inhibiting the lateral branching growth and development. In contrast, this study showed that the abscisic acid (ABA) contents increased with shading intensity. The functions of ABA in plants have been extensively investigated. It has been found to modulate lateral branching growth in response to the shading [79]. In a word, shading may affect the quantity and quality of branching by regulating the hormone level of *P. yunnanensis*. In the future, we can enhance the quantity and quality of *P. yunnanensis* lateral branching through the exogenous application of hormones.

### 3.4. The Response of Genes Related to Hormone and Photosynthesis Pathways in Pinus yunnanensis to Different Shading Levels

The transcriptome analysis is important in the research of *P. yunnanensis* molecular mechanisms [80], and many studies have successfully explored the molecular mechanisms of plant responses to shade using RNA sequencing (RNA-seq) technology [81]. Most plant photoreactions depend on the electron transport chain within the photosynthetic system. This encompasses Photosystem I (PS I), Photosystem II (PS II), the cytochrome complex, and photosynthetic electron transporters on the photosynthetic membrane [82]. The *Psb* gene family is essential for increasing the efficiency of the oxygen evolution complex within the PS II unit [83]. In the four comparison groups, our study revealed distinctive alterations in gene expression profiles for several PS II proteins, including *PsbP*, *PsbS*, *PsbR*, *PsbO*, and *Psb27*. In addition, a notable differential expression was observed in the expression of four genes involved in the regulation of PS I proteins (*PsaG*, *PsaH*, *PsaD*, *PsaL*, *PsaE*, and *PsaF*). Chlorophyll binds to the *Psa* and *Psb* subunits, forming an effective chlorophyll–protein complex that is crucial for the efficient functioning of the photosystems [84]. The downregulated expression of these genes leads to a reduction in the actual content of chlorophyll in the needles of *P. yunnanensis*, which diminishes the efficiency of light-harvesting molecules. Consequently, this decreases the effectiveness of photosynthesis in the needles. Thus, plants exhibiting shading tolerance exhibit a reduction in photosynthetic efficiency, a phenomenon that has been observed to serve as a corresponding adaptation mechanism in the context of shading stress.

A study carried out by researchers has shown that there are dynamic physiological changes in the levels of phytohormones during the growth of lateral branches [85]. For lateral branch growth, the signaling and metabolism of phytohormones are of great importance. Phytohormones are crucial in regulating the number and quality of lateral branches, primarily by controlling cell division and differentiation. This hormonal influence is integral to the developmental pathways that dictate the branching architecture of plants [71,86]. In the current study, it was demonstrated that the *SAUR* gene is significant in IAA metabolism. IAA promotes apical dominance, inhibiting the growth and development of lateral branching [87]. The expression of the *SAUR* exhibits rapid induction in response to transient changes in various environmental factors and abiotic stress [88]. Thus, the present study hypothesizes that *SAUR* might inhibit the branching growth in *P. yunnanensis* by impeding IAA transport in response to shading. In the shade-reduced lateral branching of *P. yunnanensis* growth process, the cytokinin signal transduction gene *AHP* was downregulated by shading. Prior research has indicated that the overexpression of *AHP* enhances the ability of plants to withstand environmental stress [89,90]. In a word, *AHP* was found to be expressed in a manner that was affected by shading; this reduced the adaptability of the plant *P. yunnanensis* to its environment, inhibiting the growth and development of lateral branching. However, we found that the content of ABA changed significantly under heavy shading treatment. Similarly, transcriptome-level alterations also align with this physiological phenomenon, whereby negative regulatory factors PP2C were identified within the ABA signal transduction pathway, exhibiting a downregulation in response to shading. The expression of *PP2C* was found to inhibit the growth of branches, thereby exerting a promoting effect on ABA functions [91]. In addition, *JAZ*, *TGA*, and *PR-1* expression exhibited differential expression in both the JA and SA metabolic pathways under shaded conditions. A previous study reported that exposure of plants to shading conditions, initiating the SAR, significantly increased the stability of JAZ proteins, suppressing JA-induced responses [92]. Therefore, it was postulated that the JA signal transduction pathway would be subject to regulation by shading, thereby influencing the quality of the branches. Comprehensively, shading impacted the growth of branches in *P. yunnanensis*, as well as the profiles of DEGs, influencing photosynthesis and phytohormone signal pathways. The above hormones had different effects on the growth of branches in *P. yunnanensis*. This study serves as a scientific foundation for further verification and expansion of its findings. It also provides significant importance in the revelation of the response mechanism of *P. yunnanensis* branches to shading conditions.

## 4. Material and Methods

### 4.1. Plant Material and Shading Treatment

These experiments were carried out in the nursery garden from March 2021 to December 2021. The nursery is located in a subtropical monsoon climate zone with ample sunlight and few frost and snow days. The annual *P. yunnanensis* plants after decapitation will be used as the experimental material to carry out this study under different shading conditions at the end of March. In this study, the primary variable was the intensity of the shading provided to the black clothing, categorized as follows: 0% shading or no shade, serving as the control (CK); 25% shading, categorized as mild shading (L1); 50% shading, considered moderate shading (L2); and 75% shading, classified as heavy shading (L3) (Table 1), each shading treatment group consists of 48 *P. yunnanensis* seedlings, with four replicates. All four treatments received identical field management, including timely watering and weeding, to support their growth. We collected fresh needles under various shading conditions, immediately placed them in liquid nitrogen, and subsequently stored them in a −80 °C ultralow-temperature freezer for later determination of physiological characteristic indices. These samples were collected monthly from June to December (i.e., months 6 through 12).

### 4.2. Growth Indexes and Biomass Determination

In the experiments, the number of branches and branch lengths were measured monthly on each treatment. At the end of the experiment, the fresh weight of various organs (including roots, stems, lateral branches, needles, and shoot branches) of *P. yunnanensis* seedlings under different shading treatments was measured to determine the biomass of the seedlings. Then, the *P. yunnanensis* seedlings were subjected to oven drying at 80 °C until a constant weight was achieved, after which they were weighed using an analytical balance.

### 4.3. Assay of Physiological Index Contents

We used the needles to determine the physiological indexes. By the manufacturer’s instructions, we measured the contents of soluble sugar, soluble protein, malondialdehyde (MDA), and the activities of antioxidant enzymes (catalase (CAT), superoxide dismutase (SOD), peroxidase (POD)) by using commercial kits which were bought from Suzhou Grace Biomedical Technology Co., Ltd., (Suzhou, Jiangsu, China) [93]. The specific operation steps were strictly implemented free of charge, following the instructions provided by the company. All physiological parameters were measured using the enzyme calibration. Then, we determined the chlorophyll content (Chlorophyll a, Chlorophyll b, Chlorophyll a + b, and Carotenoid) by the acetone extraction method [94].

### 4.4. Evaluation of Endogenous Hormone Content

We used the fresh branches of *P. yunnanensis* under different shading treatments in December as samples. All tissue samples were collected and frozen in liquid nitrogen, then stored in a −80 °C freezer. Six independent biological replicates were set for this study. PANOMIX Biomedical Technology Co., Ltd. (Suzhou, China) assessed endogenous phytohormone levels. The experiment data collection system primarily consists of an ultra-high-performance liquid chromatography (UHPLC) system (Vanquish, Thermo, Waltham, MA, USA) and a high-resolution mass spectrometer (Q Exactive, Thermo, Waltham, MA, USA) (https://www.thermofisher.com/, accessed on 10 March 2024). Data were acquired on the Q-Exactive mass spectrometer using Xcalibur 4.1 software (Thermo Scientific) and processed using TraceFinder™ 4.1 Clinical software. The quantified data were then exported to Excel format for further analysis [95].

### 4.5. Transcriptome Sequencing Analysis

The samples used for transcriptome analysis are the same as those used for phytohormone determination. RNA-seq of all tissue samples was conducted by PANOMIX Biomedical Technology Co., Ltd. (Suzhou, China). Trizol reagent (Invitrogen Life Technologies) was used to isolate total RNA. The RNA’s concentration, quality, and integrity were then determined using a NanoDrop spectrophotometer. We purified the library fragments using the AMPure XP system (Beckman Coulter, Beverly, CA, USA). We purified products using a Bioanalyzer 2100 system and quantified them using the Agilent high-sensitivity DNA assay. The sequencing library was then sequenced on NovaSeq 6000 platform (Illumina). Samples are sequenced on the platform to produce image files, which the sequencing platform software transforms to produce the original data in FASTQ format (raw data). We employed Cutadapt (v1.15) to filter and obtain high-quality sequences (clean data), which were subsequently assembled into transcripts using Trinity software (2.6.6). All transcripts were aligned against six databases (including NR, GO, KEGG, eggnog, Swiss-Prot, and Pfam). Gene expression levels were quantified using RSEM (http://deweylab.github.io/RSEM/, accessed on 10 March 2024) (RNA-Seq by Expectation-Maximization). Then, the DEGs were analyzed using DESeq (1.30.0) with the following screened conditions (|log2FoldChange| > 1, significant *p*-value < 0.05) [96]. We carried out GO annotation with BLAST2GO software (https://www.blast2go.com/). The Kyoto Encyclopedia of Genes and Genomes (KEGG) was used to identify significant enrichment pathways (http://www.genome.jp/kegg/, accessed on 10 March 2024). RNA-seq raw data have been uploaded to the NCBI database (accession number: PJNA1097002).

### 4.6. RT-qPCR Validation of DEGs

We selected six DEGs from the transcriptome to validate their expression levels. Primer3Plus was used to design the primers (https://www.primer3plus.com) and synthesized by Shanghai Shenggong Biology Co., Ltd., Shanghai, China (Appendix A). Taq Pro Universal SYBR qPCR Master Mix (Vazyme, China) was used to assess gene expression patterns, and the Rotor-Gene Q 5plex platform (QIAGEN, Hilden, Germany) was used for evaluation. The RT-qPCR program included an initial 95 °C for 2 min, followed by 40 cycles of 95 °C for 5 s, 60 °C for 5 s, and 72 °C for 25 s. The gene (pyTUBA1) was used as an internal control, and each reaction contained three biological replicates. The 2^−∆∆Ct^ method was used to determine gene expression levels [97].

### 4.7. Statistical Analysis

Microsoft Excel (2016) is the software of choice for the compilation of preliminary statistics. All statistical analysis was conducted using IBM SPSS Statistics (SPSS 26.0. Chicago, IL, USA), with one-way analysis of variance (ANOVA) with Duncan’s multiple-range tests (*p* < 0.05) being used as a post hoc analysis for comparison of the means. Data were plotted using GraphPad Prism 9.51 software. Pearson correlation and principal component analyses (PCA) were performed using R 4.2.3 software.

## 5. Conclusions

To sum up, shading inhibited the growth and development of branches in *P. yunnanensis*. Under the shaded condition, the quantity and quality of branches were inhibited, and the biomass allocation to each organ of *P. yunnanensis* gradually decreased with the increase in shading. Conversely, moderate shading could boost the accumulation of chlorophyll content in the branch of *P. yunnanensis*, but heavy shading produces precisely the opposite results. In addition, the physiological characteristics and phytohormones of lateral branching also had different response levels to shading. Comparison of transcriptomic data from branches under different shading conditions revealed significant changes in the expression of genes involved in photosynthesis and phytohormonal signal transduction pathways. These results establish a strong link between shading and branch growth in *P. yunnanensis* seedlings, affecting their physiological and molecular properties. Consequently, this study elucidates the control of lateral branching growth in response to shading and provides resources for investigating the mechanism in *P. yunnanensis*. However, further research is needed to validate these findings.

## Figures and Tables

**Figure 1 plants-13-01588-f001:**
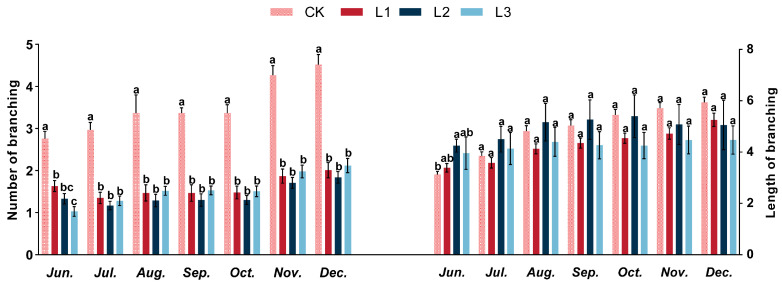
The response of *P. yunnanensis* branches number and branch length to different shading treatments. Different letters indicate significant differences between treatments within the same month, according to Duncan’s multiple range test (*p* < 0.05).

**Figure 2 plants-13-01588-f002:**
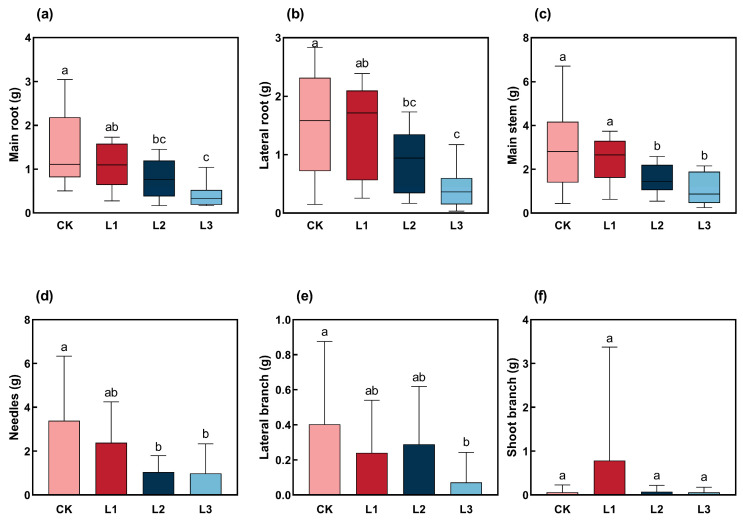
Biomass in different organs of *P. yunnanensis* responding to shading ((**a**) Main root biomass; (**b**) Lateral root biomass; (**c**) Main stem biomass; (**d**) Needles biomass; (**e**) Lateral branch biomass; (**f**) Shoot branch biomass). The same letters above the error bars indicate that there are no statistically significant differences, according to Duncan’s multiple range test (*p* < 0.05). Different colors indicate different treatments.

**Figure 3 plants-13-01588-f003:**
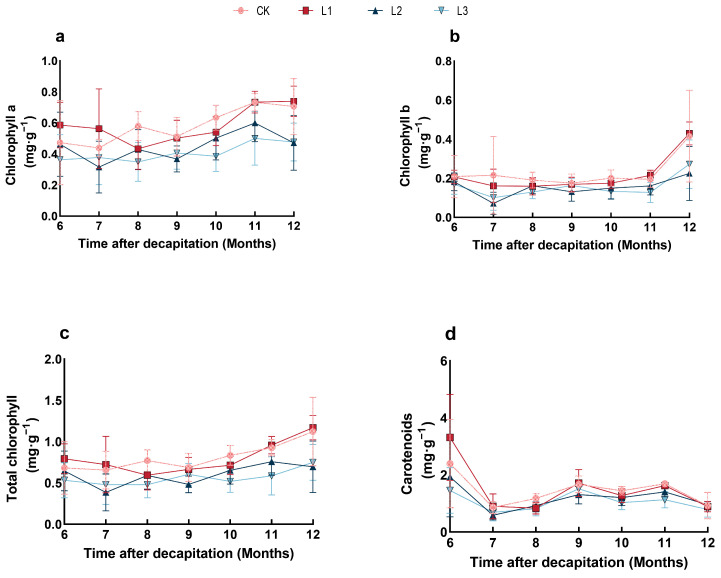
The effect of shading on chlorophyll a, chlorophyll b, total chlorophyll, and carotenoid content in branching of *P. yunnanensis* ((**a**) Chlorophyll a content; (**b**) Chlorophyll b content; (**c**) Total chlorophyll content; (**d**) Carotenoids content).

**Figure 4 plants-13-01588-f004:**
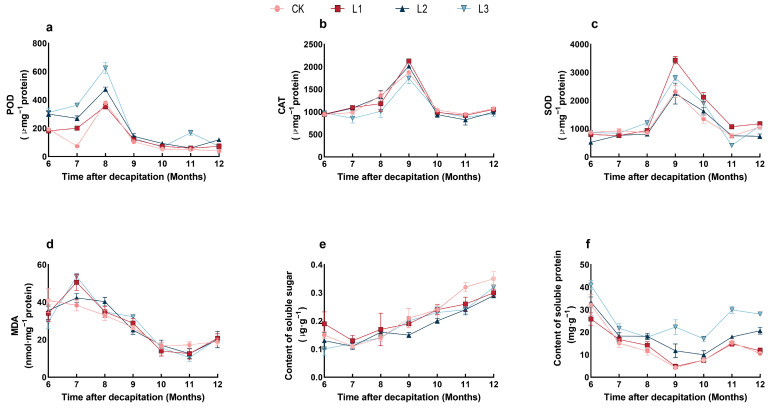
The response of physiological characteristics to different shading treatments over time. (**a**) Peroxidase (POD) content; (**b**) catalase (CAT) content; (**c**) superoxide dismutase (SOD) content; (**d**) malondialdehyde (MDA) content; (**e**) Soluble sugar content; (**f**) Soluble protein content.

**Figure 5 plants-13-01588-f005:**
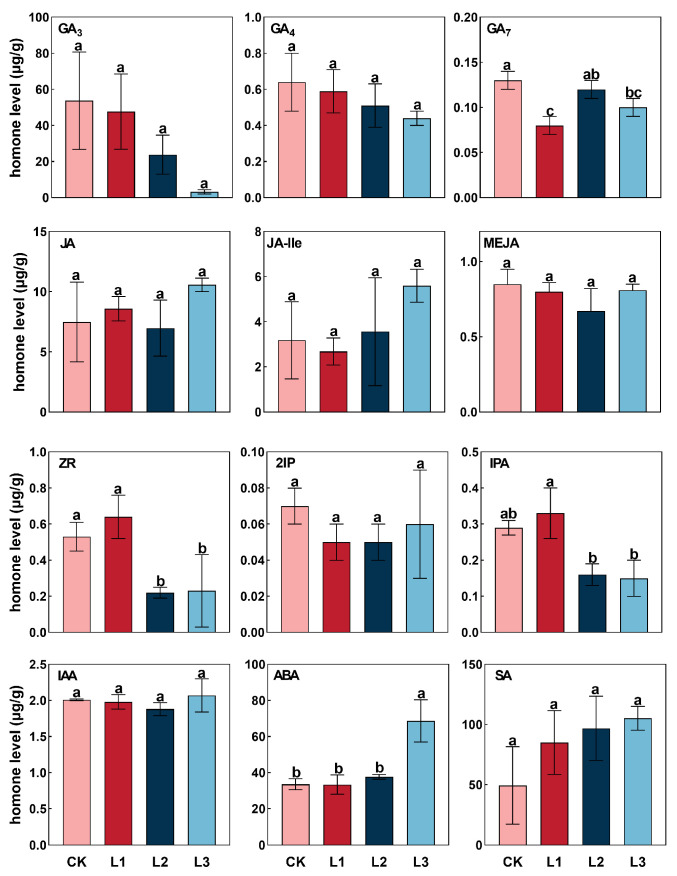
The response of endogenous hormones in the branches of *P. yunnanensis* to different degrees of shading (μg/g). Gibberellins (GA3, GA4, and GA7), auxin (IAA), abscisic acid (ABA), jasmonic acids (JA, JA-Ile, and MEJA), cytokinins (ZR, 2IP, and IPA), and salicylic acid (SA) content under different shading treatments. The same letters above the error bars indicate that there are no statistically significant differences according to Duncan’s multiple range test (*p* < 0.05). Different colors indicate different treatments.

**Figure 6 plants-13-01588-f006:**
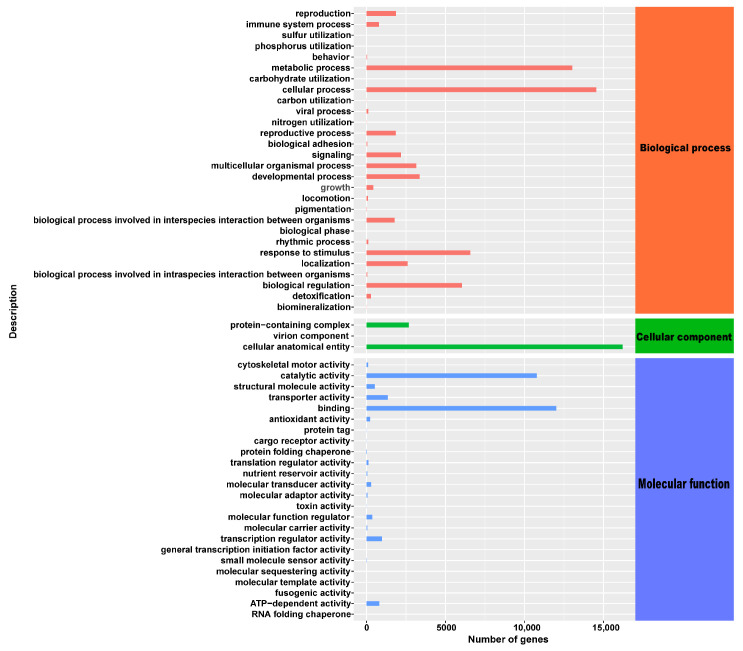
GO classifications of assembly of uni-genes in *P. yunnanensis* under different shadings.

**Figure 7 plants-13-01588-f007:**
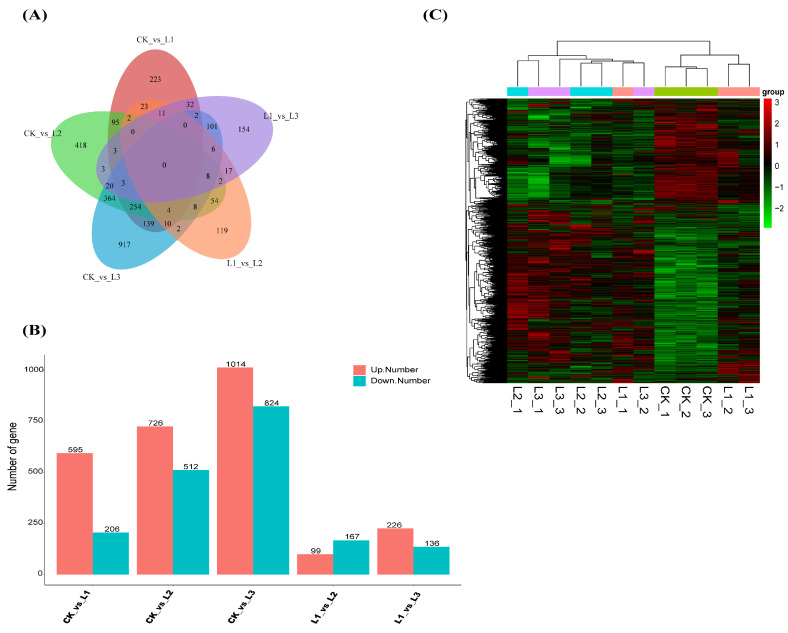
In order to characterize the DEGs, the pairs of different shading levels (CK, L1, L2, and L3) of *P. yunnanensis* seedlings were compared. (**A**) Venn diagram of DEGs in different comparison groups. (**B**) Statistical data on upregulation and downregulation of genes between different shading treatment groups. (**C**) Cluster analysis of DEGs in *P. yunnanensis* with different levels of shading.

**Figure 8 plants-13-01588-f008:**
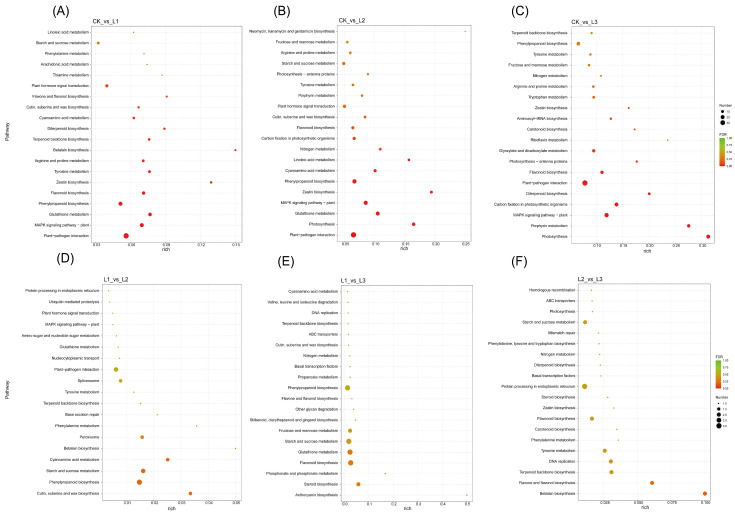
The top 20 KEGG pathways significantly enriched in differentially expressed genes under various shading conditions in *P. yunnanensis*. (**A**) CK vs. L1. (**B**) CK vs. L2. (**C**) CK vs. L3. (**D**) L1 vs. L2. (**E**) L1 vs. L3. (**F**) L2 vs. L3.

**Figure 9 plants-13-01588-f009:**
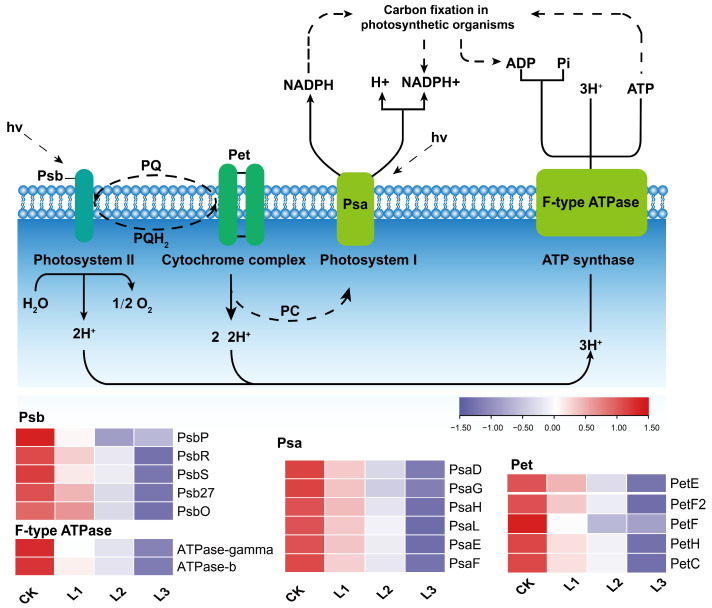
Heating maps of gene expression associated with photosynthesis pathway in *P. yunnanensis* under different shading treatments. Red colors represent up-regulation, and purple colors represent downregulation.

**Figure 10 plants-13-01588-f010:**
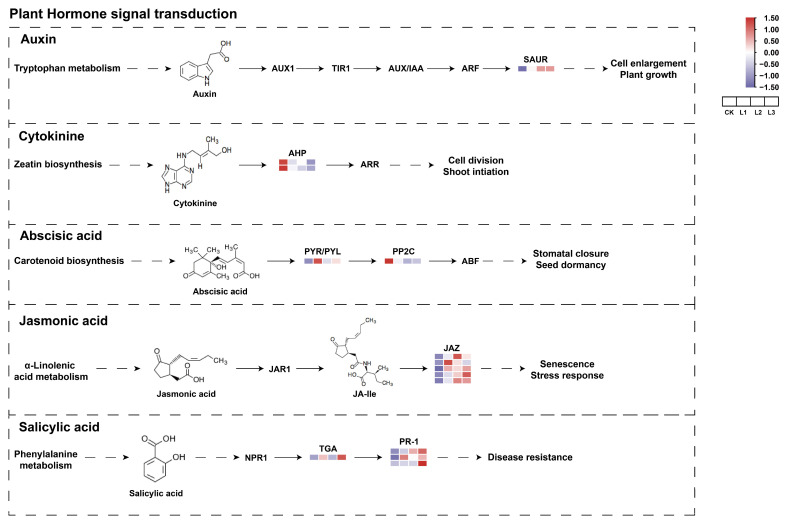
Expression profiles of gene expression related to plant hormone signal transduction pathways. Red colors present upregulation, and purple colors present downregulation.

**Figure 11 plants-13-01588-f011:**
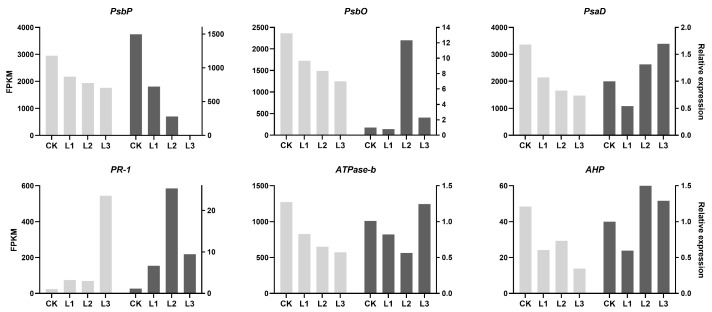
RT-qPCR validation of DEGs in *P. yunnanensis* under different shading levels. *X*-axis: shading treatments; left *Y*-axis: FPKM values from RNA-seq; right *Y*-axis: relative expression levels from RT-qPCR. The white bars represent the RNA-seq expression levels under different treatments, while the gray bars represent the RT-qPCR expression levels under different treatments.

**Table 1 plants-13-01588-t001:** The shading specifications under four different shading treatments.

Treatments	Shade Cloth Specification	Shade Level (%)
CK	None	0
L1	Black polyethylene net curtains (1 layer 2 stitches)	25
L2	Black polyethylene net curtains (1 layer 3 stitches)	50
L3	Black polyethylene net curtains (1 layer 4 stitches)	75

## Data Availability

All data generated or analyzed during this study are included in this article. All data during this study are available from the first author or request.

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
