# Peer review of "Physiological Characteristics and Transcriptomic Responses of Pinus yunnanensis Lateral Branching to Different Shading Environments"

_plants, 2024, doi:10.3390/plants13121588_

Round 1

Reviewer 1 Report

Comments and Suggestions for Authors

The manuscript with the title “Physiological characteristics and transcriptomic responses of Pinus yunnanensis lateral branching to different shading stresses” is very well written and interesting.

Row 17 and row 21 Both sentences start with „However”, please find other way to express the information, also the abstract seems too long, you can let only the important information a summary, please summarize the information in the abstract. Rows 34-36 represent discussions, please delete the information.

Row 49: “In addition, Research has found that the growth of lateral branching is influenced by”. Please write research with lowercase, it is not a name.

 Row 64: “shade avoidance syndrome (SAS).” Missing citation

Row 70: CO2 please correct the formatting

Row 152: „From Figure 2, the main and lateral root biomass decreased with the degree of shading after decapitation.” Please reformulate in such a way to cite the figure in the text at the end of the sentence. You are writing the results with the help of the figures, you are not describing the figures in the article. Also check the other similar sections.

I could not find a correlation table of values in the manuscript, however many sections are written like a correlation was performed between parameters. Please see rows 204-2019.

Row 599: Please provide a source for the methods used, for all of them.

Reviewer 2 Report

Comments and Suggestions for Authors

General recommendations and questions

Title to different shading stresses” different stresses? I think it's better - shading stress

Abstract

Line 17. “However, asexual propagation can be used to ensure the stability of good traits in the mother tree,”- in this case, are you interested in ensuring the good qualities of the mother tree or the seedling?

Asexual propagation – there are different methods of asexual or vegetative propagation (cuttings, layering, division, and budding/grafting etc). It should be specified more precisely.

Why exactly shaded environment was studied - is it in nurseries, is it used when propagating; are there problems, challenges with lighting, that's the method.... This is important to understand why the study was conducted.

Keywords: Repeating the same terms in the title and keywords is not recommended. This reduces the chances of finding the article in search engines. I would recommend using at least common name of Pinus yunnanensis in keywords.

Introduction

The theoretical part of the introduction is too long. It should be more focused and targeted, avoiding repetition.

On the other hand, there is very little information about the tree species included in the study, about its propagation, and the problems that have to be faced when preparing planting material. So it's not entirely clear why you studied shading specifically. Explain more about the relevance of the study, why it would be important, about the problems to be solved.

Results

Figure 1. The significance of the differences is indicated within the month?

Line 162-163. “The biomass of needles declined with the degree of shading, and CK treatment was significantly higher than L2 and L3 and had no correlation with L1.” Why should there be any correlation?

Line 172. “Chlorophyll is crucial in light absorption during photosynthesis.” The results do not need to contain theoretical information, nothing needs to be discussed. This applies throughout the article (e.g. Line 197-198, 221-223.). It can be moved to the Discussion section.

2.5. Response of endogenous hormone to shading treatments

All abbreviations must be given their full name the first time they are used. This applies throughout the article

Discussion

In general, it is desirable to make the Discussion section not so long, to focus more on the purpose of the study, the problem to be solved. Repetition and some jumping from one result to another and back should also be avoided. A wide range of scientific literature has been used in the preparation of the article, which is to be welcomed, however, this is not a review article, so more summarisation is recommended.

The Discussion of the results obtained in this experiment should be strictly separated from the results of other authors' studies. The quote clearly describes the problem - the question arises - was this an experiment with nitrogen, and only further on we can guess that it is about another study.

Line 376-380. “The results found that the number of lateral branching of P. yunnanensis decreases gradually with the intensification of shading under different shading treatments, the number of lateral branching in the control group (CK) was significantly greater than that in the mild shading (L1), moderate shading (L2), and heavy shading (L3) respectively. Research has indicated that shading can enhance the response efficiency of leaf growth to nitrogen doses, but it can also cause a reduction in lateral branching.”

“3.2. Response of light and pigmentation and physiological characteristics of P. yunnanensis to shading.” - very awkward title, needs to be changed.

Line 415. “Chlorophyll a is strongly related to photosynthesis and increases levels of chlorophyll and net photosynthetic rate (Pn)” What exactly did you mean by that?

Materials and methods

It is recommended to briefly describe the control conditions, whether the seedlings were in full sun, whether the weather at the study site is sunny on most days, or whether the other conditions (soil nutrient status, moisture, fertilization were the same and optimal).

Biomass determination – it would be more correct to give precise information. You determined the mass of roots, branches, and needles separately, as could be understood from the results.

4.3. Assay of physiological index contents

“The exact enzyme calibration measured all the parameters” – what does it mean?

Then, we determined the chlorophyll content (Chl a, Chl b, Chl a+b, and Caro) by the acetone extraction method.”- the reference is necessary.

4.4. Evaluation of endogenous hormone content

We used the fresh branches of P. yunnanensis under different shading treatments in December as samples” - after determining the biomass?

 Concluding remarks. In general, the article is interesting, contains new knowledge and rich data material. However, to improve the quality of the article, it is necessary to provide more accurate information about the relevance of the research. It is also recommended to shorten the Introduction and Discussion parts and make them more targeted. It is desirable to correct the errors in English here and there.

I recommend accepting this article in Plants after minor revision.
